# NAC blocks Cystatin C amyloid complex aggregation in a cell system and in skin of HCCAA patients

Michael E. March [1,8], Alvaro Gutierrez-Uzquiza[1,2,8], Asbjorg Osk Snorradottir[3,4], Leticia S. Matsuoka[1], Noelia Fonseca Balvis[2], Thorgeir Gestsson[4,5], Kenny Nguyen[1], Patrick M. A. Sleiman[1,6], Charlly Kao[1], Helgi J. Isaksson[3], Birkir Thor Bragason[7], Elias Olafsson[4,5], Astridur Palsdottir[7] & Hakon Hakonarson [1,4,6]✉

Hereditary cystatin C amyloid angiopathy is a dominantly inherited disease caused by a leucine to glutamine variant of human cystatin C (hCC). L68Q-hCC forms amyloid deposits in brain arteries associated with micro-infarcts, leading ultimately to paralysis, dementia and death in young adults. To evaluate the ability of molecules to interfere with aggregation of hCC while informing about cellular toxicity, we generated cells that produce and secrete WT and L68Q-hCC and have detected high-molecular weight complexes formed from the mutant protein. Incubations of either lysate or supernatant containing L68Q-hCC with reducing agents glutathione or N-acetyl-cysteine (NAC) breaks oligomers into monomers. Six L68Q-hCC carriers taking NAC had skin biopsies obtained to determine if hCC deposits were reduced following NAC treatment. Remarkably, ~50–90% reduction of L68Q-hCC staining was observed in five of the treated carriers suggesting that L68Q-hCC is a clinical target for reducing agents.

[1] The Center for Applied Genomics, The Children's Hospital of Philadelphia, Philadelphia, PA, USA. [2] Department of Biochemistry and Molecular Biology, Complutense Univeristy of Madrid, Madrid, Spain. [3] Department of Pathology, Landspitali University Hospital, Reykjavik, Iceland. [4] Faculty of Medicine, University of Iceland, Reykjavik, Iceland. [5] Department of Neurology, Landspitali University Hospital, Reykjavik, Iceland. [6] Divisions of Human Genetics and Pulmonary Medicine, Department of Pediatrics, The Perelman School of Medicine, University of Pennsylvania, Philadelphia, PA, USA. [7] Institute for Experimental Pathology at Keldur, University of Iceland, Reykjavik, Iceland. [8] These authors contributed equally: Michael E. March, Alvaro Gutierrez-Uzquiza. ✉email: hakonarson@chop.edu

Hereditary cystatin C amyloid angiopathy (HCCAA) is a dominantly inherited disease caused by a leucine 68 to glutamine variant of human cystatin C (hCC, L68Q-hCC)[1]. HCCAA is classified as a cerebral amyloid angiopathy (CAA), a group of diseases in which amyloid deposits form on the walls of blood vessels in the central nervous system (CNS). Although HCCAA is rightly classified as a CAA disorder due to its strong cerebral presentation, hCC deposition is systemic and also found in other internal organs. Most carriers of the mutation suffer microinfarcts and brain hemorrhages in their twenties, leading to paralysis, dementia, and death in young adults, with an average life expectancy of 30 years[2–6]. Postmortem studies in humans show hCC deposits in all brain areas, most prominently in arteries and arterioles.

Human cystatin C, a cysteine protease inhibitor that belongs to the cystatin superfamily, is a secretory type 2 cystatin, expressed in all nucleated human cells and found in all tissues and body fluids and at particularly high concentrations in cerebrospinal fluid[2,7–9]. hCC inhibits cysteine proteases like papain and legumain by its interaction through multiple binding motifs resulting from the characteristic hCC fold[9–11]. Its normal conformation is composed of a polypeptide that folds into a five-stranded β-sheet, which partially wraps around a central α-helix. The N-terminal segment and two hairpin loops build the edge of the protein, which binds into the active site of cysteine proteases and blocks their proteolytic activity[12–14]. Mutation of leucine 68 to glutamic acid destabilizes the packing between the beta sheets and the alpha helix, allowing the molecule to open. Two such open hCC molecules can interact with each other, with the helix of each molecule interacting with the beta sheet of the other; the resulting dimer is said to be the product of domain swapping[15–17]. Additionally, through a process called propagated domain swapping, long chains of molecules can be built, in which the free domain of each molecule interacts with a new hCC monomer[18]. The aggregation of proteins leads to the formation of highly ordered pathogenic fibrillar aggregates, called amyloid fibrils[19,20], which are implicated not only in HCCAA but also in a wide range of neurodegenerative diseases such as Alzheimer's, Parkinson's, Creutzfeldt–Jacob's, and Huntington's disease[20].

The degree of amyloid maturation observed in cystatin C deposits has been shown to vary between tissues (i.e., less prominent maturation in skin than in brain)[21]. Although deposits in the skin are not composed of amyloid fibers, quantitative studies on hCC deposition within the skin of mutant carriers showed that symptomatic carriers had significantly higher levels of hCC immunoreactivity in their skin than asymptomatic carriers. The fact that the quantity of hCC deposition in skin was associated with the progression of the disease in the CNS shows that skin biopsies are useful in assessing disease progression and could, therefore, be of use in the evaluation of therapeutic interventions[22].

Protein oligomers of different pathogenic amyloidogenic proteins precede the fibril formation stage in HCCAA and other diseases, although for HCCAA, it is unclear if such oligomers lead directly to pathogenic fibrils or if assembly of fibrils occurs most rapidly from monomers[23]. Drugs reducing aggregation of amyloid-producing proteins have the potential to reduce the formation of toxic oligomers known to occur in several types of amyloidosis[24,25]. Previous investigations have suggested that preventing domain swapping of hCC might be used for treatment of HCCAA[24]; Nilsson et al.[26] developed variants of WT hCC and L68Q-hCC with intra-chain-stabilizing disulfide bonds preventing domain swapping; stabilized molecules could not form dimers. These results suggest that the knowledge of the molecular mechanism causing the transition of physiologically normal and soluble proteins to toxic oligomers and insoluble fibrils is essential for the development of treatment strategies.

Östner et al.[24] have previously attempted to prevent polymerization of hCC monomers, or disrupt or remove multimeric species, through various approaches. As mentioned, use of stabilized hCC monomers demonstrated that preventing domain swapping prevents aggregations. Antibodies can be raised specifically against the domain swapped, dimeric form of hCC; those antibodies were able to specifically remove dimers of hCC, and not monomers, from patient plasma using size-exclusion chromatography[27]. A high-throughput screen of compounds has been pursued using the US Drug Collection (composed of 1040 FDA-approved compounds, http://www.msdiscovery.com/usdrug.html) in an effort to find molecules that prevent dimerization[24]. Although promising, this approach required large amounts of purified hCC protein produced in bacteria, and the compounds identified as inhibiting dimer formation were for the most part used at concentrations too high to be considered therapeutic in an organism.

To create a system in which to test the ability of a compound to impact hCC multimerization while gaining some insight into its toxicity, we created cell lines that express high amounts of either wild type or mutant hCC. This study describes our characterization of the cell lines and the monomeric and multimeric hCC that they create and attempts to nontoxically interfere with aggregation of the mutant protein. We also present the effects of NAC taken as a drug prescribed for mucolytic therapy or taken voluntarily as a supplement on hCC deposition in skin biopsies of human subjects with HCCAA.

## Results

**Incubation with glutathione impairs hCC di-/oligomerization in cellular extracts and supernatants.** In our efforts to create a cell model and detection system for examining the effects of potential therapeutic agents on aggregation of L68Q-hCC, we created stable transfectants of HEK293T cells overexpressing and secreting either WT- or L68Q-hCC. We observed profound laddering on Western blots in lysates and the existence of only high-molecular-weight species (HMW) in supernatants of cells expressing the mutant (Supplementary Fig. 1), but only if the samples were run under nonreducing conditions. The laddering in lysates corresponds to the predicted molecular weights of monomeric hCC and discrete multimers of hCC (dimer, trimer, and tetramer), which we refer to as smaller species (dimer, trimer, and tetramer) and low-molecular-weight (LMW) complexes. Conversely, wild-type protein expressed and secreted predominantly as a monomer. Treatment of mutant lysates or supernatants with the reducing agent DTT (dithiothreitol) collapsed practically all the protein to monomeric size, suggesting that disulfide bonds, either intramolecular or intermolecular, are required for observation of multimers and larger species on denaturing sodium dodecyl sulfate polyacrylamide gel electrophoresis (SDS-PAGE).

DTT is not appropriate for therapeutic use, but multiple naturally occurring reducing agents exit. To examine the efficacy of such a compound, supernatants and cellular extracts of WT or L68Q-hCC-expressing cells were treated with different concentrations of reduced glutathione (GSH) at 37 °C for 15 min. As Fig. 1 shows, treatments with 3 or 10 mM of GSH severely reduced the amount of dimer and/or HMW oligomer observed in both the secreted and the intracellular fraction of L68Q variant. Quantitation showed that 3 mM of GSH displayed an ≈90% reduction of the HMW in the secreted fraction and an ≈50% inhibition in the intracellular fraction (lysates) of the L68Q-hCC variant (Fig. 1 and Supplementary Fig. 2). Unexpectedly, we observed an overall increase in the amount of protein observed with increasing amounts of GSH. For the mutant protein, this

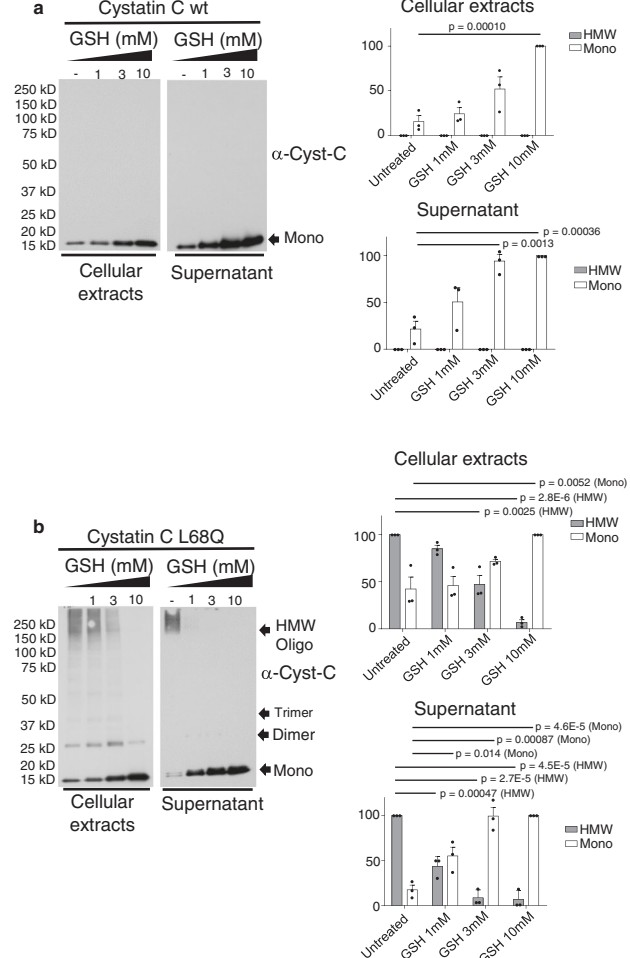

**Fig. 1 Incubation with glutathione impairs cystatin C di-/oligomerization in cellular extracts and supernatants.** Supernatants and cellular extracts from hCC WT- (**a**) or L68Q-expressing (**b**) cells were incubated in the presence of glutathione (GSH) at the indicated concentrations for 1 h at 37 °C. Samples were mixed with 2% SDS without reducing agents prior to electrophoresis, and protein levels were detected by Western blot for cystatin C, and amounts of monomeric hCC (Mono) or high-molecular-weight hCC species (HMW) were quantitated ($N = 3$ independent experiments; one-tailed student $t$ test; indicated $p$ values were less than 0.05, without correction for multiple testing, with respect to untreated). In blots and graphs, bars are means, error bars represent standard deviation. Gel images for replicate experiments are shown in Supplementary Fig. 2.

apparent increase may be explained by the destruction of large protein complexes resulting in increasing amounts of monomer. However, this explanation would not be expected to apply to the wild type. A second potential explanation is that the epitope recognized by the anti-cystatin C antibody used for blotting may be partially masked under nonreducing conditions. Treatment with reducing agents under such circumstances would result in increased detection by Western blot. In addition, as seen in Supplementary Fig. 1B, the wild-type protein can be observed in higher-molecular-weight species in longer exposures; those larger species would also serve as sources of monomer upon treatment with reducing agents.

**Incubation with NAC or gluthathione impairs dimerization of secreted L68Q-hCC.** The oxidized/reduced glutathione pair is critical to fight against oxidative stress, and, as shown in Fig. 2, it can effectively disrupt the dimers and HMW oligomers of hCC.

Accordingly, we analyzed whether another reducing agent, the commonly used dietary supplement n-acetyl cysteine (NAC), would affect the oligomerization/dimerization of secreted hCC. Supernatants were treated with different concentrations of GSH and NAC at 37 °C for 60 min. As Fig. 2 shows, treatments with 3 or 10 mM of either reagent reduced the oligomerization/dimerization levels of secreted L68Q-hCC in vitro. Quantitation showed almost complete ablation of HMW with 3 mM concentrations of either GSH or NAC (Fig. 2 and Supplementary Fig. 3). This result demonstrates that GSH or NAC are able to decrease the oligomerization levels of the pathogenic version of L68Q-hCC and can be potentially used as treatment of HCCAA patients. Supplementary Figure 3 contains the replicate experiments used for quantitation, and shows the large difference in secretion between wild-type and mutant proteins.

**Detection of L68Q-hCC complexes by sucrose gradient centrifugation.** The SDS-PAGE-based method for detecting oligomers of L68Q-hCC denatures proteins as part of its execution, and as such, the method itself may contribute to breaking HMW and LMW species down to monomers. As a denaturation-independent method of detecting the effects of reducing agents on L68Q-hCC complexes, sucrose gradient-density ultracentrifugation was performed. Lysates from L68Q-hCC cells were treated with NAC or the strong reducing agent β-mercaptoethanol and loaded on top of sucrose gradients. Following ultracentrifugation, fractions were collected from the gradient and proteins were precipitated. The presence of hCC in each fraction was detected by Western blot (Fig. 3). L68Q-hCC was present in all fractions of the gradient, with the majority of the protein present in the middle- and lower-density fractions. However, detectable amounts of L68Q-hCC were also present in the densest fractions of the gradient. Treatment with β-mercaptoethanol resulted in a loss of L68Q-hCC from the two densest fractions of the gradient, suggesting the loss of the largest complexes. Treatment with NAC resulted in loss of mutant protein from the densest fraction, and a reduction of protein present in the second densest fraction. These results provide evidence, from an independent biochemical method, that NAC is capable of breaking HMW complexes of L68Q-hCC into smaller or less-dense species.

**Presence of GSH or NAC reduces oligomerization of secreted L68Q-hCC at 24, 48, and 72 h.** To investigate whether NAC or GSH reduces the oligomerization of secreted L68Q-hCC in a cellular system, we treated cells expressing WT or L68Q-hCC with both agents. Cells were seeded in plates and allowed to secrete hCC for 48 h, at which point increasing concentrations of GSH or NAC were added to the cultures. Cells were cultured for 72 h in the presence of reducing agents, with samples of the supernatants being removed after 24, 48, and 72 h. Oligomerization status of hCC was determined by Western blot at each time point. No gross differences in morphology were observed for the duration of the experiment in the presence of all concentrations (up to 10 mM) of both reducing agents. As shown in Fig. 4 (and Supplementary Fig. 4), treatment of cells with 10 mM GSH or NAC completely abolished the presence of HMW and LMW at the 24 and 48-h time points, and appreciable but incomplete reduction of HMW and LMW persisted at 72 h. Treatments with lower doses of NAC or GSH were only incompletely effective at 24 and 48 h and no significant effect was detected after 72 h (Fig. 4).

**Incubation with NAC derivatives impairs dimerization of secreted L68Q-hCC.** NAC should have a variety of protective antioxidant effects; however, some clinical trials failed to confirm

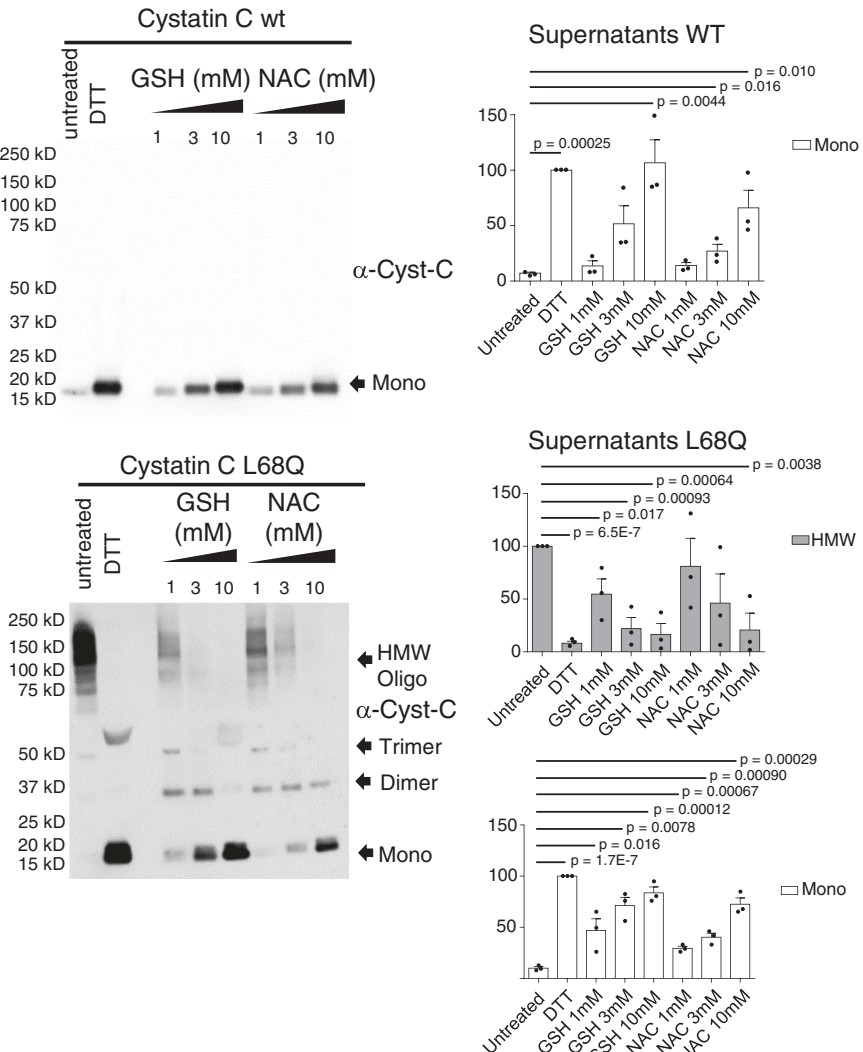

**Fig. 2 Glutathione and N-acetylcysteine impairs oligomerization of secreted cystatin C L68Q.** Supernatants were incubated in the presence of the indicated concentrations of glutathione or NAC for 1 h at 37 °C. Samples were mixed with 2% SDS without reducing agents prior to electrophoresis, and protein levels were detected by anti-cystatin C antibody and amounts of monomeric hCC (Mono) or high-molecular- weight hCC species (HMW) were quantitated. No HMW fraction was detected in supernatants from HEK-293T cells stably expressing hCC WT ($N = 3$ independent experiments; one-tailed student $t$ test; indicated $p$ values were less than 0.05, without correction for multiple testing, with respect to untreated). Bars are means, error bars represent standard deviation. Gel images for replicate experiments are shown in Supplementary Fig. 3.

such beneficial effects. A possible explanation for this absence of beneficial effects could be its low oral bioavailability or its complex intestinal absorption. Additionally, several discrepancies have been reported in the literature about the ability of the NAC to cross blood–brain barrier (BBB). These problems have stimulated the search for alternative pharmacologically favorable cysteine pro-drugs like NAC amide (NACA) or NAC-Methyl ester (NAC-Me). These newly prepared derivatives of NAC include groups to facilitate membrane diffusion and have been shown to cross the BBB and to protect from oxidative injury in animal models and in cultured cells[28,29]. In order to investigate if these agents display similar effects to that of NAC, we analyzed whether other reducing agents (like NACA and NAC-Me) would affect the oligomerization/dimerization of secreted hCC. Supernatants were treated with different concentrations of NAC, NACA, or NAC-Me at 37 °C for 60 min. As Fig. 5a shows, treatments with 3 or 10 mM of NAC or 1, 3, or 10 mM of NACA or NAC-Me reduced the oligomerization/dimerization levels of secreted L68Q-hCC in vitro. Quantitation showed almost complete ablation of HMW with 3 mM concentrations of NAC or concentration of 1 mM of NACA or NAC-

Me. This result demonstrates that NAC or its derivatives (NACA and NAC-Me) are able to decrease the oligomerization levels of the pathogenic version of L68Q-hCC; indeed, NAC derivatives showed a stronger activity (around 10 times stronger activity). We also treated cells for 24 h with the NAC derivatives, as in Fig. 4. As shown in Supplementary Fig. 6, the derivatives were more effective than NAC in cell culture, and as with NAC, no gross differences in morphology were observed for the duration of the experiment in the presence of all concentrations (up to 10 mM) of all reducing agents. Supplementary Figure 5 contains the replicate experiments used for quantitation, and shows the large difference in effect between NAC and its derivatives.

**Effects of NAC administration in HCCAA patients.** The L68Q-hCC mutation is a founder mutation in the Icelandic population that occurred approximately 18 generations ago during the 16th century[4], with most mutation carriers suffering microinfarcts and brain hemorrhages as young adults, ultimately leading to paralysis, dementia, and death. Given the results from the

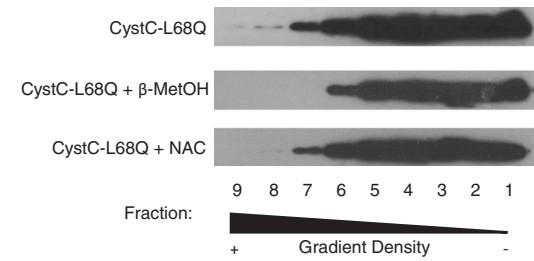

**Fig. 3 N-acetylcysteine reduces L68Q-hCC density in sucrose-gradient ultracentrifugation.** L68Q-hCC cell lysates were treated with β-mercaptoethanol (β-MetOH) or n-acetyl cysteine (NAC) for 1 h at 37 °C. Lysates were layered on top of sucrose step gradients, ranging from 5% sucrose at the top (fraction 1) to 30% at the bottom (fraction 9) in 5% increments. Gradients were ultracentifuged in a swinging bucket rotor for 16 h at 4 °C and 246,000×*g*. Fractions were collected from gradients, proteins were precipitated with tricholoroacetic acid, and precipitated proteins were dissolved in PBS before analysis by SDS-PAGE under reducing conditions. Gels were blotted for cystatin C. The image is representative of three independent experiments and sucrose gradients.

described cellular work, it became evident that NAC could potentially be useful to treat this devastating disease. The proband in one of the HCCAA families in Iceland had sustained three serious strokes over 9 months at the age of 22 and required an intensive care unit stay due to paralysis and respiratory failure complicated by recurrent atelectasis. Following the third stroke, the proband was bound to a respirator and was suffering from mucus plugging and atelectasis. Coincidentally to our findings in the cell models, the proband was treated with the mucolytic agent NAC and mucous plugging gradually resolved. As a result of the effects of NAC in the described cell system, the proband was maintained on the NAC therapy. A parent and a sibling of the proband, both carriers of the mutation, also elected to begin taking NAC as a food supplement in light of the effects of NAC observed in the cell system (Fig. 6a). This was prior to subsequently enrolling in a clinical trial testing the efficacy of NAC in patients with HCCAA, that launched in 2019. Three additional extended family members and carriers of the mutation (Fig. 6b) also decided to begin taking NAC as a food supplement, based on information from the proband family, prior to enrolling into the NAC clinical trial in 2019. All three individuals had a historical skin biopsy at the time of their diagnosis and a follow-up skin biopsy done 6–24 months after they began taking NAC as a food supplement.

While the amount of mature amyloid fibrils differs between tissues in HCCAA patients, it has been recently shown that the amount of deposits in the skin correlates with the symptomatic or asymptomatic status of patients, with higher deposition in skin observed in symptomatic individuals[22]. While we cannot directly monitor the impact of NAC on hCC deposition in the brain, we chose to investigate the effect of NAC therapy on hCC deposition in skin as a proxy for its activity in the brain. Skin biopsies from subjects were obtained under a research protocol described in the "Methods" section. For the first three patients, three biopsies were taken from each individual (Fig. 6a). Biopsy #1 was a historical biopsy performed at the time of diagnosis of the proband, over 5 years ago for research purposes; biopsy #2 was performed after the proband was being treated with 400 mg of NAC 4× per day for a total duration of 9 months, for the respiratory problems resulting from mucus plugging as discussed above; this was just prior to all three subjects starting a NAC dose of 600 mg 3× per day and served as a baseline to assess for NAC biomarker response in the parent and sibling. Biopsy #3 was performed

following 6 months of NAC treatment at 600 mg 3× per day and was used to determine skin biomarker response to NAC therapy. All biopsies were stained together and at the same time to limit batch effects. On the three additional patients, 2 biopsies were taken from each individual (Fig. 6b). Biopsy #1 was a historical biopsy performed in 2017 (carrier 1), 2018 (carrier 2), and 2015 (carrier 3), for research purposes. Biopsy #2 was performed after carrier 1 had been treated with 600 mg of NAC 3× per day for 24 months, carrier 2 for 8 months, and carrier 3 for 9 months.

Figure 6 demonstrates the changes in staining over the course of the observations. The cystatin C deposition was most evident in the basement membrane between the epidermis and dermis (the deposition has been shown to begin in this area in asymptomatic carriers), but was also present in basement membranes around dermal arteries, arterioles, veins, hair follicles, sebaceous glands, fat/sweat glands, and arrector pili muscles. Overall, the drug reduced the percentage of the dermis and epidermis showing hCC staining. Whether decreased deposition in skin following treatment correlates with decreased deposition in brain or other tissues or organs will require more detailed and sophisticated investigation. In addition, deposition in the skin has been shown to not consist of mature amyloid fibers, although amyloid-like threads have been shown by electron microscopy analyses[22,30]. Whether or not NAC treatment affects fully mature amyloid deposits found in brain and other tissues remains to be investigated.

In the first patient set (Fig. 6a), the proband had very high levels of hCC protein complex deposition in the first skin biopsy. Subsequent biopsies reveal the proband had not progressed in terms of deposition; rather, the deposition had decreased by about 40%, as measured by hCC immunoreactivity between skin biopsies #1 and #2 during which the proband was on NAC therapy 400 mg 3× per day to treat lung atelectasis. In contrast, the sibling (who also carries the L68Q mutation and was not on NAC between biopsies #1 and #2) showed significant progression in immunoreactivity, reflective of an increase in hCC protein complex deposition in the skin. The parent showed smaller progression of deposition between biopsies #1 and #2. When skin biopsy #3 was obtained 6 months later, the proband demonstrated a visible reduction in the hCC deposition in biopsy #3 in comparison with the original skin biopsy. In the second patient set (Fig. 6b), carriers 1 and 2 showed a visible reduction in the hCC deposition, while carrier 3 showed a slight progression. It should be noted that carrier 3 (Fig. 6b, bottom panel) had first biopsy done over 3 years before beginning NAC therapy, suggesting that the intensity of staining in the skin may have been higher at the time NAC therapy began and that a potential response was masked by the time lag.

The reduction in hCC deposition in the proband from first biopsy to third (approximately 15 months at two different doses) was estimated at ~70% (Fig. 6a). The parent demonstrated an ~50% reduction in staining on biopsy #3 (Fig. 6a), and the sibling ~30% after 6 months of NAC. In the second cohort, carriers 1 and 2 showed a visible reduction in the hCC deposition with carrier 1 having near- complete clearance following 600 mg of NAC 3× per day for 24 months. Carrier 3 showed an increase in staining; as discussed, this analysis is complicated by the gap between the baseline biopsy and initiation of therapy. In total, five of six treated subjects showed a demonstrable reduction in hCC staining in skin following NAC therapy. The pre- and post-treatment stainings for all six subjects show statistically significant reduction (one-tailed, paired student *t* test, *p* = 0.028, Fig. 6c).

## Discussion

Identification of agents with the ability to reduce L68Q-hCC dimerization and amyloid fibril formation is the key for the

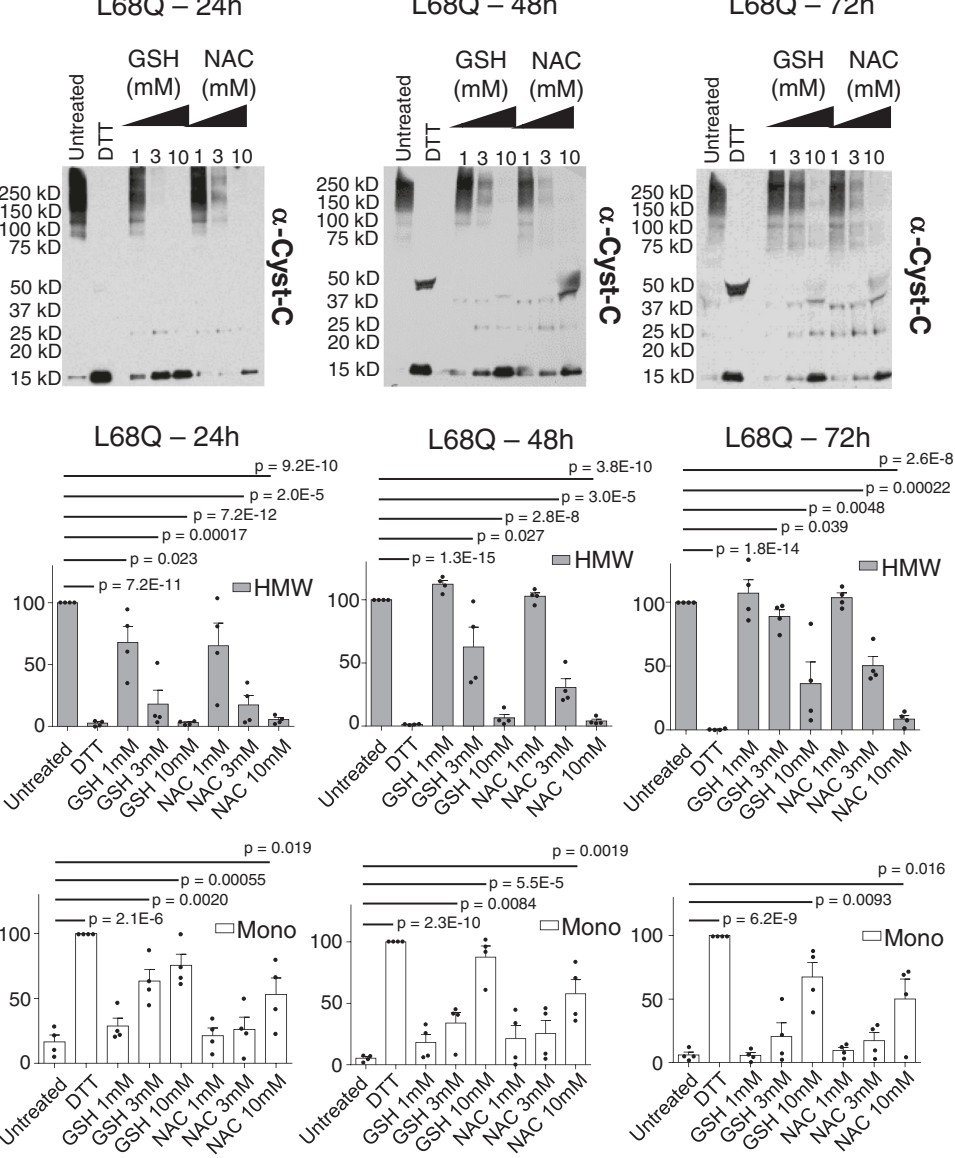

**Fig. 4 NAC impairs oligomerization of secreted hCC L68Q.** 293T cells expressing WT or L68Q cystatin C were incubated with media containing the indicated amount of either glutathione (GSH) or n-acetyl cysteine (NAC) for 24, 48, or 72 h. Small amounts of supernatant were removed from the cells and analyzed by Western blot at the indicated times. On days 2 and 3, only the supernatants from cells expressing hCC L68Q variant were analyzed. Samples were mixed with 2% SDS without reducing agents prior to electrophoresis, protein levels were detected by anti-cystatin C antibody, and amounts of monomeric hCC (Mono) or high-molecular-weight hCC species (HMW) were quantitated (N = 4 independent transfections; one-tailed student t test; indicated p values were less than 0.05, without correction for multiple testing, with respect to untreated). Bars are means, error bars represent standard deviation. Experiments from which replicate measurements are derived are shown in Supplementary Fig. 4.

development of treatments for HCCAA. Here, we created cells that produce and secrete detectable levels of hCC (WT or L68Q) capable of oligomerizing under nonreducing conditions, and we then show that short incubation with GSH, NAC, or NAC derivatives breaks oligomers into monomers. We show that treatment of cell cultures with either GSH or NAC reduces oligomerization of the secreted hCC L68Q at 24, 48, and 72 h. In addition, we have successfully tested NAC derivatives NAC-A or NAC-Me, which are also able to reduce oligomerization of the secreted hCC L68Q. Treatment with NAC in human patients not only prevents ongoing deposition of cystatin C protein complex in the skin, but also reduces prior deposits in a significant way. Statistically significant reduction of staining in skin was seen in the six patients treated. The small number of subjects in this study is a caveat, and analysis of a larger number of subjects in the context of a clinical

trial would be required to confirm these results and assess efficacy of the compounds. A NAC clinical trial was launched in 2019 and the results are anticipated in Q2, 2021 (EudraCT#: 2017-004776-56). Previous systems for the study of the dimerization of hCC had been developed. Most of the in vitro work was performed mainly with bacterially produced and purified WT hCC[24]. Absence of tests of promising candidates on the L68Q variant makes it difficult to translate the in vitro results obtained by previous analysis into in vivo studies.

Previous cellular studies have made use of exogenously expressed hCC, both wild type and mutant, in characterization of intracellular trafficking, secretion, multimer formation, and protein stability. Merz et al.[31] observed a large hCC-containing complex of greater than 70 kDa in lysates of transfected Chinese hamster ovary cells. The results presented in much of the

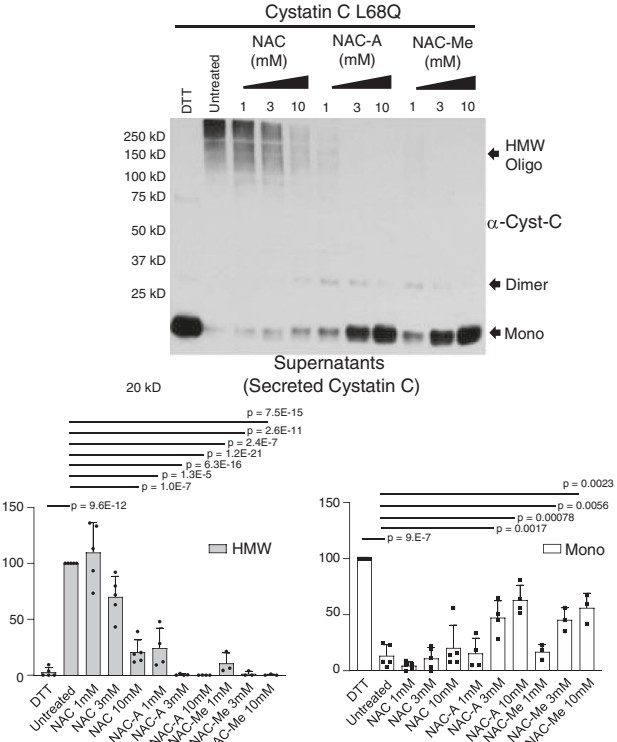

**Fig. 5 NAC derivatives impairs oligomerization of secreted hCC L68Q.**
Supernatants were incubated in the presence of the indicated concentrations of n-acetyl cysteine (NAC), NAC amide (NACA), or NAC-Methyl ester (NAC-Me) for 1 h. Samples were mixed with 2% SDS without reducing agents prior to electrophoresis, protein levels were detected by anti-cystatin C antibody, and amounts of monomeric hCC (Mono) or high-molecular-weight hCC species (HMW) were quantitated (N = 5 independent experiments, 2 containing only NAC-A and 3 containing NAC-A and NAC-Me; one-tailed student t test; indicated p values were less than 0.05, without correction for multiple testing, with respect to untreated). Bars are means, error bars represent standard deviation. Gel images for replicate experiments are shown in Supplementary Fig. 5.

previous work dealing with secreted protein have focused on comparatively small protein species (below 50 kDa). To our knowledge, this work is the first to describe the presence of L68Q-specific high-molecular-weight species in supernatants of transfected cells secreting hCC.

Exposure of L68Q-hCC expressing cells to GSH, NAC, or NAC derivatives in culture led to reduced accumulation of high-molecular-weight species in supernatants. Reducing agents might only be effective on the extracellular pools of L68Q-hCC in vivo, but this may be sufficient to be of therapeutic benefit. Pathogenesis in HCCAA occurs as the result of accumulation of protein extracellularly, particularly inside blood vessels, where the aggregates would be accessible. In addition, NAC has been shown to cross the BBB[32,33], but with limited bioavailability in the brain. It has been previously proposed that derivatives of NAC may possess better properties in this regard[28,29]. We have tested two NAC derivatives in our cell-based system, both of which appear to have improved pharmacokinetic profiles[28,29,34,35] and they have shown even stronger effects than NAC, suggesting they may be therapeutic options. More testing needs to be performed before these derivative compounds can be administered to humans.

L68Q-hCC is highly amyloidogenic, and subjects carrying the corresponding mutation suffer from severe cerebral amyloidosis, leading to brain hemorrhage and death in early adult life[26]. The

age of onset of clinical symptoms has dropped dramatically over the natural history of the disease, with a reduction in lifespan from approximately 65 years of age in 1825 to the current day average of 30[4]. Lifestyle changes that occurred in the Icelandic population along with the economic and industrial changes of the 19th century are postulated as environmental factors influencing the onset of disease in HCCAA. Increased consumption of carbohydrates occurred over this period. Hyperglycemia has been linked to oxidative stress in diabetes[36,37], and it is possible that increased carbohydrates in Icelandic diets in the 19th century created enough oxidative stress in L68Q-hCC carriers to make presentation of the disease more severe. This hypothesis is compatible with our results; increased oxidative stress would increase multimerization of L68Q-hCC, and dietary reducing agents like NAC would reverse that effect and prevent deposition of new aggregates. Therapeutic benefits would be derived mainly from the lack of new occlusions causing new strokes.

Under normal conditions, GSH levels are regulated by the rates of its synthesis and its export from cells. However, GSH levels are also influenced by agents or conditions that alter the thiol redox state (leading to the formation of glutathione S-conjugates or complexes) or that disrupt the distribution of GSH among various intracellular organelles. In addition, GSH levels are affected by nutritional status and hormonal/stress levels, exhibit developmental and diurnal variations, and are affected by certain physiological states, including pregnancy and exercise[38–45]. Physiological levels of GSH in blood should provide an appropriate antioxidant environment that avoids extracellular accumulation of proteins. However, the presence of mutations like L68Q-hCC or deficiencies in the levels of GSH, as a consequence of nutritional status or age, could lead to the undesired accumulation of misfolded proteins[4]. Treatments that lead to increases in GSH may help combat oxidative stress-induced damage, which the presented results suggest may include HCCAA-associated amyloid deposition.

NAC is a synthetic N-acetyl derivative of the endogenous amino acid L-cysteine, a precursor of the antioxidant glutathione. Both GSH and NAC already have been approved for use in humans and have been administered at high doses for long periods without adverse side effects[46,47]. They work as a direct reactive oxygen species scavenger and as a source of sulfhydryl groups. NAC, in addition, also regenerates liver stores of GSH. These effects confer to NAC the ability to reduce disulfide bonds and are the reason why NAC is widely used to reduce viscosity and elasticity of mucus. Our data show that treatment with strong reducing agents (DTT) or antioxidants (GSH and NAC) abolishes hCC oligomerization in our cell model system. This effect indicates that disulfide bond formation is essential for the oligomerization process. However, the precise mechanisms by which GSH or NAC modulates hCC protein oligomerization remain elusive and need to be fully explored. Disulfide bonds have not been shown to be directly involved in the dimerization process[26]; however, the presence of two disulfide bonds in human cystatin C (as in all type 2 cystatins), and the preservation of them in the dimeric structure indicates its key role in the protein structure[26]. We postulate that the intramolecular disulfide bonds are essential for the correct folding of the hCC monomer and for the exchange of three-dimensional subdomains between the two subunits of the dimer and their reduction abolishes the oligomerization. Alternatively, the L68Q proteins in our system may form intermolecular disulfide bonds that are not described in the existing dimeric crystal structures. Further work will be required to characterize the precise complexes that are being produced and created in this system. Despite these limitations, this cell system led to identification of a compound (NAC) that has effects on cystatin C deposition in the skin in preliminary work with patients.

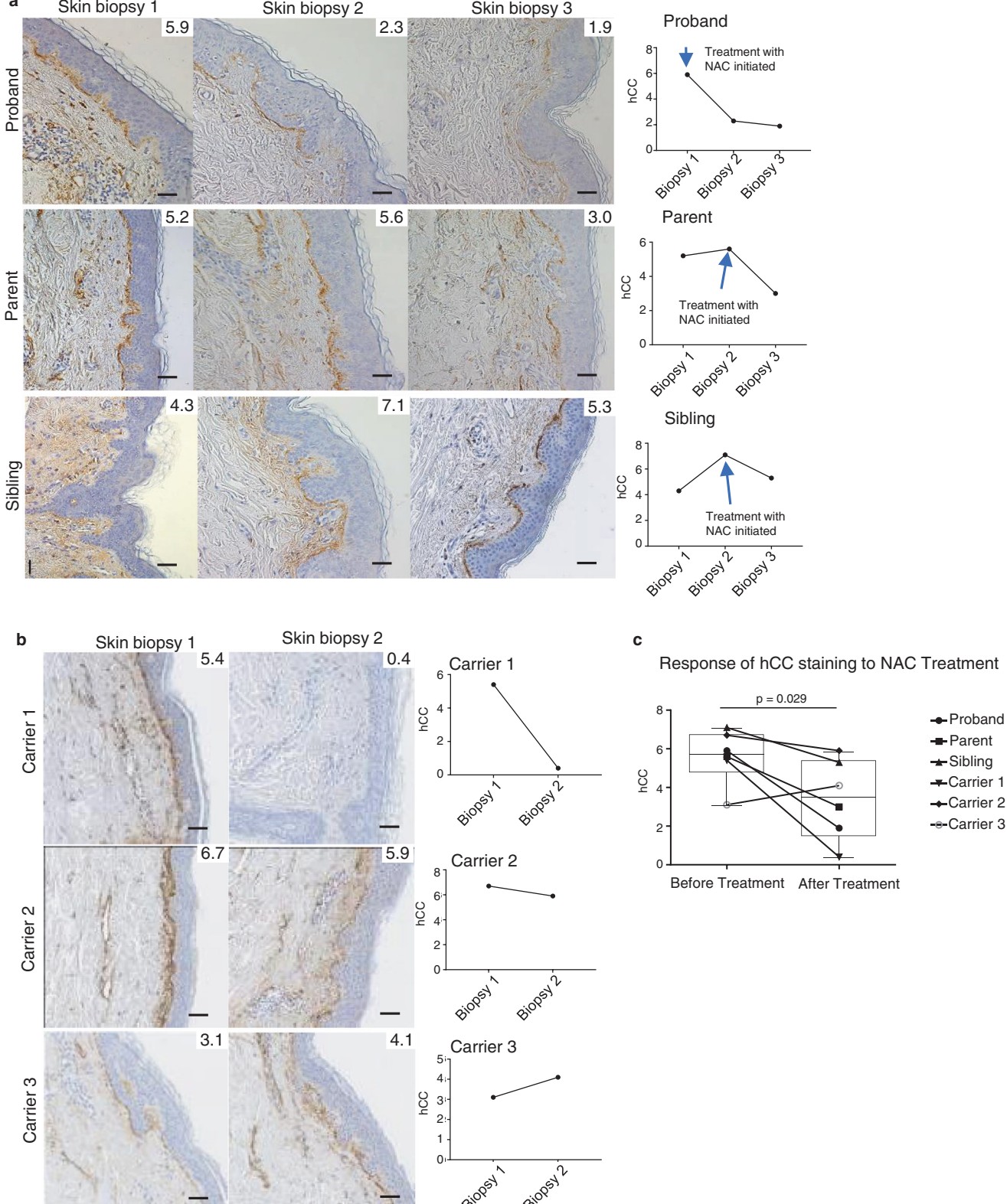

We hypothesize that treatment with NAC will reduce oligomerization of secreted hCC, thus reducing amyloid formation in the brain of patients with HCCAA, either directly or through the induction of GSH. Treatment with GSH may be effective; however, its low bioavailability limits the interest of supplementation as therapeutic option for the treatment of patients with HCCAA.

NAC appears to be a good candidate because of its role in restoring GSH levels, antioxidant properties, and its ability to break disulfide bonds[48]. In addition, NAC supplementation significantly improved coronary and peripheral vasodilatation[49]. Specific to brain disorders, NAC has been trialed with some efficacy in patients with Alzheimer's disease[50], and our data

**Fig. 6 Effects of NAC administration in HCCAA patients.** Cystatin C immunostaining (brown stain) was performed on skin biopsies from two separate cohorts of carriers of the hCC L68Q variant, using a rabbit-anti human cystatin C antibody. **a** Skin biopsy 1 was obtained when the family participated in research, approximately 18 months prior to the proband beginning NAC therapy for respiratory issues at this time. Skin biopsy 2 was obtained after proband had been treated with NAC for 9 months, at which time the parent and sibling began therapy. Skin biopsies 3 were obtained after 6 months of therapy with NAC (parent/sibling) and 15 months total by the proband. A marked reduction was seen in the proband following the entire course of therapy, and the parent and sibling after 6 months of NAC therapy. **b** Skin biopsy 1 was obtained approximately 2 (carrier 1), 1 (carrier 2), or 4 (carrier 3) years ago, respectively. Skin biopsy 2 was taken following 24 months (carrier 1), and approximately 8 months (carriers 2 and 3) of 600 mg of NAC 3× per day. Numbers in the corner of each image represent the percentage of pixels within the dermis and epidermis that are covered by hCC staining. In **a**, **b**, a single-biopsy section was analyzed per subject per time point. Scale bars are 50 mm in all images. **c** Boxplot represents the quantification by densitometry of the images before and after the treatment (one-tailed, paired student $t$ test; treated vs. untreated—$p = 0.028$, number of subjects = 6). The bottom and top bounds of the box represent the first and third quartiles, respectively, the center line represents the median, and whiskers extend to the extreme-most points. Data for each subject are represented by the indicated points, and points for subjects before and after treatment are connected by lines.

suggest that it warrants further investigation for effectiveness in HCCAA. It is worth noting that antioxidants in general, and in some cases NAC specifically, have been clinically trialed in other dementia-related settings with minimal or no significant effect[51]. Endpoints like memory and comprehension may represent damage to the brain that is irreversible, leading to disappointment in previous trials. Here, we propose that the ability of NAC to prevent new depositions of mutant hCC will have a protective effect on carriers, acting to prevent new strokes from occurring. While effects on dementia may be minimal, benefits in life extension could be considerable.

The reduction observed in the cystatin C protein complex deposition in the skin biopsies of L68Q-hCC carriers with NAC indicates that NAC can have an effect on hCC aggregation or deposition in patients when taken orally. As cystatin C deposition is systemic in HCCAA patients, it is possible that precipitation and accumulation of amyloid in the brain and blood vessels is reduced when reduction is observed in the skin. Currently, there are no noninvasive techniques for observing hCC accumulation in and around capillaries in the brain, limiting our ability to directly assess effectiveness. However, apart from what is discussed below, no new events have occurred in any of the subjects given NAC, all of whom have continued therapy. The proband is now approximately over five years post the third and last major stroke. A minor bleed was observed in the brain of the proband approximately 12 months ago; however, this bleed surprisingly spontaneously resolved within 48 h with no clinical effect, which is very atypical for the natural history of this devastating disease where recurrent bleeds tend to become more severe. Following on these observations, a clinical trial has started to evaluate the effect of NAC therapy on HCCAA patients (EudraCT#: 2017-004776-56).

We feel there may be a relationship between HCCAA and other cerebral amyloid angiopathies like Alzheimer's disease that is analogous to that of familial combined hypercholesterolemia (FCH) and general hypercholesterolemia. Statin drugs were developed to treat familial FCH (patients with FCH develop stroke and myocardial infarction in their 20 s); it subsequently became evident that elevated cholesterol was harmful and a major risk factor for myocardial infarction and stroke, and that patients with cardiovascular risk factors benefitted from statin treatment. HCCAA is an enhanced amyloid precipitation that occurs in early life and leads to catastrophic events in the 20s and early dementia. This process of protein deposition is somewhat comparable, but occurs slower in Alzheimer´s disease, leading to dementia much later in life. If the underlying mechanisms of protein deposition and pathogenesis are sufficiently similar, similar or identical treatments may be effective.

In summary, this study provides evidence suggesting that mutant hCC is a pharmacological target for reducing agents like NAC, and provides evidence that oral administration of NAC can affect hCC deposition in the skin of patients with HCCAA. A larger clinical trial is underway to determine if there is evidence of other systemic benefits from NAC intake in patients with HCCAA (EudraCT#: 2017-004776-56).

## Methods

**Cells and hCC WT vs. L68Q variant expression constructs**. Human embryonic kidney 293 (HEK-239T) cells were obtained from ATCC (Manassas, Virginia) and grown at 37 °C in Dulbecco's modified Eagle's medium (DMEM) supplemented with 10% fetal bovine serum. A plasmid containing a cDNA of *CST3* was obtained from Dharmacon (Lafayette, CO). The full-length coding sequence was amplified with a c-terminal Myc tag by PCR and ligated into the *Eco*RI and *Not*I sites of pBABE-CMV-Puro[52]. The L68Q mutation was introduced by QuikChange site-directed mutagenesis. Primer sequences are listed in Supplementary Table 1. All sequences were confirmed by Sanger sequencing. Wild-type and mutant constructs were transfected into HEK-293T cells using FuGENE HD (Promega, Madison, WI), with 3 μg of DNA and 9 μl of the transfection reagent, according to the manufacturer's protocol. After transfection, cells were incubated with fresh medium containing puromycin (1 μg/ml) for 3 weeks. After selection, stable clones of each transfectant were generated by limiting dilution. Clones were screened by Western blot using anti-hCystatin C antibody MAB1196 (R&D, Minneapolis, MN).

**Western blot**. HEK-293T cells expressing hCC WT or the L68Q variant were washed twice with ice-cold phosphate-buffered saline (PBS) and lysed on ice using a freshly prepared ice-cold cell lysis buffer containing 50 mM Tris-HCl, pH 7.4, 100 mM NaCl, 50 mM β-glycerophosphate, 10% glycerol (w/v), 1% NP-40 (w/v), 1 mM EDTA, 2 mM NaVO4, and a complete, EDTA-free, protein inhibitor cocktail (Roche Applied Science, Mannheim Germany) at 20 μl per mL of lysis buffer. After clearing the cell lysates by centrifugation (10 min, 21,000×*g*, 4 °C), the supernatants were collected and used for Western blotting. Sample buffer containing SDS, glycerol, Tris-HCl pH 6.8, and bromophenol blue was added to each sample to the following final concentrations: 2% SDS, 10% glycerol, 50 mM Tris-HCl, and 0.02% bromophenol blue. In samples that were reduced, either DTT (50 mM final concentration) or β-mercaptoethanol (5% final concentration) were added. Equal volumes of lysate or supernatant samples were loaded on NuPAGE 4–12% Bis–Tris gels (Thermo Fisher Scientific, Waltham MA) without heating/boiling. Proteins were transferred to PVDF membranes (Millipore, Billerica, MA) and blotted with anti-hCystatin C (MAB1196 [R&D, Minneapolis, MN] used at 500 ng/ml), and developed by enhanced chemiluminescence (ECL, Thermo Fisher Scientific). Western blots were acquired via enhanced chemiluminescence using either ECL film or the KwikQuant Imager from Kindle Biosciences. KwikQuant Images were processed upon acquisition using manufacturer-provided macros for Adobe Photoshop CC (20161012.r.53 ×64), and densities of bands were determined using the gel analysis features of Fiji v1.53c[53]. Graphs and statistical analysis were done in Graphpad Prism v9.0.0.

**Drug treatments**. HEK-239T cells were plated on 6-well plates and cultured for 2 days, at which point reduced glutathione (GSH) (Sigma, St. Louis, MO), N-acetylcystein (NAC) (Sigma), N-acetylcysteine amide (NACA) (Sigma), or N-acetylcysteine-Methyl (NAC-Me) (Sigma) were added to the indicated concentrations. Cells were incubated with indicated compounds for 24 h or 72 h, and 100 μl samples of supernatants were removed at 24, 48, and 72 h. Supernatants were cleared by centrifugation (10 min, 21,000×*g*, 4 °C). Sample buffer containing SDS, glycerol, Tris-HCl pH 6.8, and bromophenol blue was added to each sample to the following final concentrations: 2% SDS, 10% glycerol, 50 mM Tris-HCl, and 0.02% bromophenol blue. When indicated, cells were washed with PBS and lysed and cystatin C levels were determined by means of Western blot analysis.

**Sucrose gradient ultracentrifugation**. Lysates of L68Q-hCC untreated or treated with 10 mM NAC or 5% β-mercaptoethanol were analyzed by sucrose gradient density ultracentrifugation[54]. Briefly, cells were lysed in the lysis buffer described above, and loaded onto a stepwise sucrose gradient composed of 700-μl layers of sucrose of increasing percentage (top-to-bottom: 5%, 10%, 15%, 20%, 25%, and 30%). Gradients were ultracentrifuged for 16 h at 4 °C at 246,000×g. After centrifugation, fractions were removed from the gradients, 500 μl per fraction. Fractions were numbered 1 (top of gradient, lowest density) through 9 (bottom of gradient, highest density). Total protein from each fraction was precipitated using trichloroacetic acid, washed with acetone, and resuspended in 20 μl of PBS. Proteins were prepared for SDS-PAGE under reducing and denaturing conditions, and L68Q-hCC was detected by western blot.

**Statistics and reproducibility**. The means and standard deviations of data were calculated. One-sided $t$ test tests were used to determine the level of significance with respect to the untreated samples, with $p < 0.05$ being considered statistically significant.

For summarization and statistical analysis of western blots, data are derived from the number of experiments indicated in the figure legend. For Figs. 1, 2, 4, and 5 the replicate experiments are shown in Supplementary figures, as indicated in the figure legends. For Supplementary Fig. 1b, the experiment was repeated three times with similar results.

**Tissue samples**. All samples were processed at the Department of Pathology, Landspitali National University Hospital, Reykjavik, Iceland. Punch skin biopsies (4 mm, central back, one per individual) were obtained. All tissue samples were formalin-fixed, paraffin-embedded, and cut into 3-μm sections for immunohistochemistry.

**Immunohistochemistry**. Sections were deparaffinized and rehydrated in xylene and ethanol. They were then immunostained using the EnVision Detection System Peroxidase/DAB, Rabbit/Mouse kit (Dako, K4065). Incubations with primary antibody (anti-Cystatin C, Sigma, HPA013143, used at 1:100 dilution) were performed at room temperature for 30 min. After incubation with a primary antibody, sections were incubated with EnVision FLEX/HRP. Sections were washed between steps with Tris-buffered NaCl solution with Tween 20, pH 7.6 (Dako, S3306). All sections were incubated with 3,3′-diaminobenzidine solution (Dako, K4065) for 10 min. Sections were counterstained with haematoxylin for 5 min followed by washing with tap water for 10 min. Finally, sections were dehydrated with 100% ethanol and xylol followed by coverslipping with mounting medium (Pertex, Histolab). Images were acquired with a Nikon Eclipse 50i microscope equipped with a Nikon DS-Fil digital camera and a Nikon Digital Sight DS-U2 camera controller. Image panels were constructed using the GNU Image Manipulation Program (GIMP 2.8.10).

**Quantification of Cystatin C immunostaining in skin biopsies**. Cystatin C immunostaining in the patient biopsies was quantified by semiautomated image analysis using the ImageJ software (http://rsbweb.nih.gov/,v1.47). Bright-field images of a section from all individuals were captured on a Nikon Eclipse 50i microscope equipped with a Nikon DS-Fil digital camera and a Nikon Digital Sight DS-U2 camera controller at a resolution of 2560 × 1920 pixels using a Nikon ×4/ 0.3NA objective. RGB color images of the sections were imported to ImageJ. On each image, a rectangular 1300 × 1300-pixel region of interest (ROI) was defined. The ROI was positioned so that one edge was placed at the periphery of the epidermis ensuring that the ROI extended over the epidermis and well into the dermis. The captured RGB image was transformed to the $L^*$, $a^*$, $b^*$ color space (CIE 1976, CIELAB). The $b^*$ channel was thresholded using the automated threshold function of ImageJ and fine-adjusted to correspond to all stained areas on the section, ensuring a minimal bleed through of differentially stained structures. Using the thresholded image, the percentage area fraction covered by the threshold was calculated. The resulting number characterizes the immunostaining load in the ROI and was subsequently used to quantitatively investigate differences in the degree of immunostaining between samples[22].

**Treatment of HCCAA patients with NAC**. Three 4-mm skin biopsies were obtained from the back from each study participants. The first biopsy from the proband was obtained when she was diagnosed with HCAA prior to initiation of NAC. The second biopsy was taken after the proband had experienced over 9 months of NAC therapy (400 mg 4× per day) to treat mucus plugging in the lungs following her third stroke. The third biopsy was taken 6 months thereafter and during those six months, she was taking 600 mg 3× per day of NAC. Thus, the proband received 400 mg of NAC 4× per day for 9 months followed by 600 mg of NAC 3× per day for 6 months. The proband parent and sibling had their first biopsy taken at the same time as the proband's first biopsy prior to NAC. Their second biopsy was taken at the same time the proband had her second biopsy; this was prior to parent and sibling receiving NAC. The third biopsy was taken 6 months later after the parent and sibling received 600 mg of NAC 3× per day for 6 months.

For the second patient set, two biopsies were taken from each individual. Biopsy #1 was a historical biopsy performed in 2017 (carrier 1), 2018 (carrier 2), and 2015 (carrier 3), respectively at the time of diagnosis of HCCAA. Biopsy #2 for all three patients was obtained at the time of enrollment into the NAC clinical trial (EudraCT#: 2017-004776-56), at which time carrier 1 had taken 600 mg of NAC 3× per day for over 24 months and carrier 2 for approximately 8 months. Carrier 3 had problem with access to the drug and had taken NAC for less than 8 months at 600 mg 3× per day prior to the second biopsy, including a few stops in between.

**Study approval**. All necessary permits for the use of skin biopsies from L68Q-CST3 carriers, and records associated with samples as well as medical information, were obtained from the National Bioethics Committee of Iceland, reference numbers 04-046-S2 and 15-060-S1. The study design and conduct complied with all relevant regulations regarding the use of human study participants and was conducted in accordance to the criteria set by the Declaration of Helsinki. All subjects signed the informed consent. The NAC therapy was clinically and serendipitously prescribed as a mucolytic therapy to treat lung atelectasis in the proband. The other family members and carriers took NAC as a dietary supplement (i.e., purchasing NAC online through Amazon) based on the results from the cell-based assays, while waiting to formally enroll into a clinical trial that was designed to assess the efficacy of NAC in patients with HCCAA and launched in 2019. The NAC supplement used by the participants was N-acetylcystein (NAC) developed by NOW Health Group Inc., IL, USA (available on Amazon).

**Reporting summary**. Further information on research design is available in the Nature Research Reporting Summary linked to this article.

## Data availability

All uncropped images from western blots are presented in Supplementary Data. Raw data from patient biopsies in Fig. 6 will be made available upon request to the corresponding author. We note that HCCAA is a very rare medical condition and only 23 individuals in Iceland have been diagnosed with the condition. As such, the data are sensitive and will be deidentified and potentially masked further prior to honoring requests for access to raw data to protect the identity of the study participants. Source data are provided with this paper.

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

## Acknowledgements

We thank all the patients involved in this study for their participation. Funding for this work was provided by an Institutional Development Fund to the Center for Applied Genomics from Children's Hospital of Philadelphia and a sponsored Research agreement with Artic Therapeutics LLC. Funding was provided to Dr. Gutierrez-Uzquiza from Autonomous Community of Madrid (CAM). Spain. "2017-T1/BMD-5468" 2018-2020.-IP: Alvaro Gutierrez Uzquiza.

## Author contributions

M.M.: Conception or design of the experimental work, data collection, data analysis and interpretation, drafting the article, critical revision of the article, and final approval of the version to be published. A.G.U.: Conception or design of the experimental work, data collection, data analysis and interpretation, drafting the article, critical revision of the article, and final approval of the version to be published. A.O.S.: Data generation, including processing of skin biopsies, data analysis and interpretation, critical revision of the article, and final approval of the version to be published. L.M.: Data generation, analysis and interpretation, critical revision of the article, and final approval of the version to be published. N.F.B.: Data generation, analysis and interpretation, critical revision of the article and final approval of the version to be published. T.G.: Data generation, critical revision of the article, and final approval of the version to be published. K.N.: Data analysis and interpretation, critical revision of the article, and final approval of the version to be published. P.S.: Data analysis and interpretation, critical revision of the article, and final approval of the version to be published. C.K.: Data analysis and interpretation, critical revision of the article, and final approval of the version to be published. H.I.: Contribution of patient material, data analysis and interpretation, critical revision of the article, and final approval of the version to be published. B.T.B.: Contribution of patient material and data analysis and interpretation, critical revision of the article, and final approval of the version to be published. E.O.: Contribution of patient material, data analysis and interpretation, critical revision of the article, and final approval of the version to be published. A.P.: Contribution of patient material, data analysis and interpretation, critical revision of the article, and final approval of the version to be published. H.H.: Conceived and designed the study, supervised all aspects of the study, drafting the article, critical revision of the article, and final approval of the version to be published.

## Competing interests

The authors declare the following competing interests. Drs Hakonarson and Kao are co-founders of Arctic Therapeutics LLC, which is funding the study. The remaining authors declare no competing interests.
