## [Peer Review File · Nature Communications]

Reviewers' Comments:

Reviewer #1:

Remarks to the Author:

Hereditary cystatin C amyloid angiopathy was first described by Arni Arnason as a dominantly inherited disorder of cerebral haemorrhage with a high prevalence in Iceland in his thesis "Apoplexie und ihre Verebung" published in 1935. His final major conclusions and wishes, based upon his extensive clinical experience, were that future research should make it possible to diagnose the disorder, before the patients were afflicted by cerebral haemorrhages and to develop useful treatment options. Development of a diagnostic procedure took 53 years involving great efforts from inter alia some of the authors of the present manuscript. As reported in the manuscript, the basic disorder-producing event is a mutation in a cystatin C allele, which means that a variant of cystatin C is produced with a glutamine residue replacing a leucine residue at position 68 of the single polypeptide chain of cystatin C. Since then large efforts have been devoted to understand how this mutation can produce amyloid deposits of cystatin C and how this process might be stopped or delayed. However, these efforts have not involved cellular and animal studies, which have been difficult to establish for the disease.

The present study is so far unique in the field, since it uses both relevant cellular systems and patients to test if reducing substances might delay, or stop, the deposition of cystatin C amyloid in vivo. The experiments are carefully performed and the results support the conclusion of the manuscript that treatment with certain reducing agents might be beneficial for the patients. The results of the work, and the interpretation of them, agree with previous knowledge in the field gathered by in vitro research. The relevant previous work is given in the selected references.

I have only found one error in the manuscript. The text "leucine 68 to glutamic acid" should be "leucine 68 to glutamine".

I recommend publication of this well executed study, with results that might fulfil the second wish of Arni Arnason.

Yours sincerely,
Anders Grubb

Reviewer #2:

None

Reviewer #3:

Remarks to the Author:

The manuscript by March and colleagues describe reducing agent, NAC, inhibition of cystatin-C aggregation in vitro and suggest that it may function in vivo in patients with HCCAA patients. The use of reducing agents as an anti-aggregant treatment for amyloid forming peptides and proteins has been extensively studied. The development of cell lines expressing wild type and mutant cystatin-C to improve the speed of identification of potential treatments is useful however is not a novel approach. The manuscript utilizes a single technique to demonstrate the efficacy of NAC to inhibit both wild type and mutant cystatin-C, as a proof of concept for drug development the use of a single technique is not sufficient, in vitro efficacy should be demonstrated in at least an additional 2 techniques. The in vivo skin biopsy data presented from two patients treated with NAC are tantalizing, however, are not sufficiently powered with respect to patient numbers and biopsy sampling. In light of these concerns, there can be no direct cause-effect relationship determined from the data as presented.

Specific Concerns:

1. The western blot system that was utilized for the presence and absence of oligomers/various molecular weight species was designed to be used under denaturing conditions. Thus, the technique to show inhibition of aggregation will also contribute to the reduction of protein aggregates. This needs to be reconciled.

2. Although acknowledged in the manuscript, there is no indication that inhibition of aggregation has any positive effects on the cell lines.
3. The western blot data, both primary figures and biological replicates, are inconsistent with respect to the exposure times and the presence/absence of non-specific bands even when utilizing the same reagents. For example, Figure 1 the monomer bands are overexposed and thus will generate inaccurate quantification as well as the contribution of a single or multiple bands to the designated "monomer". In contrast, Figure 2 monomers are adequately exposed however without an extended exposure there is no way to reconcile the total loss of the high molecular weight species. This needs to be reconciled.
4. There is inadequate information presented on the biopsy samples to be able to fully derive how the data was generated. There is no indication of the number of biopsies per patient, number of sections analysed per biopsy and registration of the skin biopsies across time and patient.

Reviewer #4:

Remarks to the Author:

The authors developed a cell-based model to investigate potential therapeutic approaches for hereditary cystatin-C amyloid angiopathy (Islandic type, L68Q), a devastating disease that leads to microvascular infarcts, hemorrhages and dementia at a young age. The authors generated cells expressing mutated cystatin-C forming aggregates and performed a drug screen to identify compounds that would prevent the aggregation. Glutathione (GSH) and N-acetylcysteine (NAC) were found to potently inhibit the formation of the aggregates. Remarkably, administration of NAC to 2 patients with the disease resulted in reduction of the amyloid accumulation in the skin assessed 6 months later. It is suggested that antioxidants may be a promising treatment for amyloid angiopathy.

This is an interesting study providing proof of principle that treatment with NAC or derivatives could prevent the damaging consequence of cystatin-C aggregation in brain vessels. The following issues deserve the attention of the authors and should also least be discussed.

Although the results in vitro are suggestive, the findings in patients are more limited due the small number examined. Furthermore, the deposition in the skin could be regionally variable and the limited samples provided may not accurately reflect treatment effects.

Another question concerns whether systemically administered NAC reaches the brain in sufficient amounts to be effective, since the most relevant pathology occurs in the brain, not easily accessible due to the blood-brain barrier.

Antioxidants have been tested in Alzheimer's disease, as in many other diseases in which radicals have been implicated, without efficacy. Based on these previous failures, the enthusiasm for using antioxidants has been diminished and this issue warrants some mention.

The accumulation of the aggregates in the skin does not seem to involve blood vessels. Would amyloid infiltrating the vascular wall of brain vessels be removed by the treatment as easily as in the skin?

What is Glu in figure 4?

Point-by-point response to reviewer comments.

Reviewer 1:

No criticisms.

Reviewer 3:

The manuscript utilizes a single technique to demonstrate the efficacy of NAC to inhibit both wild type and mutant cystatin-C, as a proof of concept for drug development the use of a single technique is not sufficient, in vitro efficacy should be demonstrated in at least an additional 2 techniques. The in vivo skin biopsy data presented from two patients treated with NAC are tantalizing, however, are not sufficiently powered with respect to patient numbers and biopsy sampling. In light of these concerns, there can be no direct cause-effect relationship determined from the data as presented.

We have added an additional *in vitro* technique to the manuscript (sucrose gradient centrifugation) which shows beneficial impact of NAC on mutant hCC oligomers. We have added a third family member to the patient study, and improved the description of the methods and the included results.

Specific Concerns:

1. The western blot system that was utilized for the presence and absence of oligomers/various molecular weight species was designed to be used under denaturing conditions. Thus, the technique to show inhibition of aggregation will also contribute to the reduction of protein aggregates. This needs to be reconciled.

We have added an additional *in vitro* technique to the manuscript (sucrose gradient centrifugation) which shows beneficial impact of NAC on mutant hCC oligomers.

2. Although acknowledged in the manuscript, there is no indication that inhibition of aggregation has any positive effects on the cell lines.

Additional text has been added to the manuscript to highlight that the major impact of NAC treatment is dissolution of protein complexes. The pathogenic characteristic of the disease is accumulation of extracellular protein complexes; we therefore consider effects on the cells producing the hCC to be of minimal concern.

3. The western blot data, both primary figures and biological replicates, are inconsistent with respect to the exposure times and the presence/absence of non-specific bands even when utilizing the same reagents. For example, Figure 1 the monomer bands are overexposed and thus will generate inaccurate quantification as well as the contribution of a single or multiple bands to the designated "monomer". In contrast, Figure 2 monomers are adequately exposed however without an extended exposure there is no way to reconcile the total loss of the high molecular weight species. This needs to be reconciled.

We thank the reviewer for his feedback concerning the quality of the westerns shown. Secretion of wild type and mutant hCC occurs at dramatically different levels, and as such proper control of exposure times is difficult. We have replaced the western in the original Figure 1 (which is now Supplemental Figure 1) with more appropriately exposed images. Additionally, the images in Figure 3 (which is now Figure 2), where wild type and mutant, and monomers and oligomers are all directly compared in the same experiments, have been replaced with more appropriately exposed images. Additionally, the supplemental figures contain all of the exposures used to create the figure and the associated statistics, with the relevant regions of each gel highlighted. It is possible for the reader to completely evaluate relative expression levels of monomers and oligomers in both wild type and mutant supernatants.

4. *There is inadequate information presented on the biopsy samples to be able to fully derive how the data was generated. There is no indication of the number of biopsies per patient, number of sections analysed per biopsy and registration of the skin biopsies across time and patient.*

Additional description of the methods has been added to the manuscript.

Reviewer 4

Although the results in vitro are suggestive, the findings in patients are more limited due the small number examined. Furthermore, the deposition in the skin could be regionally variable and the limited samples provided may not accurately reflect treatment effects.

We have added an additional family member to the patient data, and expanded the description of the methods to increase clarity.

Another question concerns whether systemically administered NAC reaches the brain in sufficient amounts to be effective, since the most relevant pathology occurs in the brain, not easily accessible due to the blood-brain barrier.

We have added text to the manuscript to address this issue. The pathogenic effects of L68Q-hCC actually occur on the blood side of the blood-brain barrier, as the secreted mutant hCC aggregates and accumulates in blood vessels. Even if NAC does not cross the blood brain barrier, it will still be able to access the pathogenic aggregates, acting essentially as a pipe cleaner, and preventing further accumulation of aggregates and possibly dissolving existing complexes.

Antioxidants have been tested in Alzheimer's disease, as in many other diseases in which radicals have been implicated, without efficacy. Based on these previous failures, the enthusiasm for using antioxidants has been diminished and this issue warrants some mention.

Text has been added to the discussion to address this point.

The accumulation of the aggregates in the skin does not seem to involve blood vessels. Would amyloid infiltrating the vascular wall of brain vessels be removed by the treatment as easily as in the skin?

Coauthor's previous publications (especially "Pathological changes in basement membranes and dermal connective tissue of skin from patients with hereditary cystatin C amyloid angiopathy") highlights the strong correlation between amounts of deposition in the skin and deposition in the post-mortem brain. While we are currently unable to examine deposits in living brains, we believe that the effects observed in the skin indicate that NAC at the very least blocks deposition of new mutant protein, which would be beneficial in the brain even if existing complexes are resistant to dissolution. This is supported by the length of time the proband has experienced since the last large stroke event while taking NAC. While only a single patient, the natural history of HCCAA is that following events occur with increasing frequency until death following the first major stroke. We feel the proband's length of time since last event are strongly indicative of NAC's benefit.

What is Glu in figure 4?

We had incorrectly labeled reduced glutathione as GSH in some figures and Glu in others. Labeling has been fixed and is now be consistent throughout the manuscript.

Reviewers' Comments:

Reviewer #3:

Remarks to the Author:

The manuscript by March et al describe the potential use of NAC for the treatment of hereditary Cystatin-C aggregation in HCCAA patients. The in vitro data has been strengthened by the inclusion of new data, however, the small number of patients and the variable results still are a concern for translation to a larger population. The language in regards to the potential of the treatment and the use in the brain needs to be more conservative as the data do not support strong statements. Furthermore, the title needs to include prose to the effect that the NAC effects are only on the dermal lesions and not a full treatment.

Reviewer #4:

Remarks to the Author:

The authors have addressed most of the issues raised in the review. The additional patients included strengthen the findings and conclusions of the study.

However the authors need to be more cautious about NAC being able to clear the aggregates from the vascular wall. The blood-brain barrier is at the level of the endothelium, which is the inner layer of the vessel wall. If NAC works as a "pipe cleaner", as stated, then it must be assumed that the aggregates cross the endothelium from the brain side leaving the vessel wall otherwise intact. The aggregates permeate the entire vascular wall and induce considerable structural and functional damage to the vessel wall through inflammation and oxidative stress, and removing them may still leave a damaged vessel behind. On the other hand, preventive treatment with NAC may protect the vessel through a "sink effect" draining low order aggregated into the circulation and preventing their accumulation in the vessel wall and consequent damage.

REVIEWERS' COMMENTS

Reviewer #3 (Remarks to the Author):

The manuscript by March et al describe the potential use of NAC for the treatment of hereditary Cystatin-C aggregation in HCCAA patients. The in vitro data has been strengthened by the inclusion of new data, however, the small number of patients and the variable results still are a concern for translation to a larger population. The language in regards to the potential of the treatment and the use in the brain needs to be more conservative as the data do not support strong statements. Furthermore, the title needs to include prose to the effect that the NAC effects are only on the dermal lesions and not a full treatment.

We thank the reviewer for further constructive criticism. The title has been modified to reflect that the observed effects are in the skin. We have modified the body of the text to be more conservative with regards to therapeutic claims (highlighted text in lines 275-278, 358-363, 381-384 for examples)

Reviewer #4 (Remarks to the Author):

The authors have addressed most of the issues raised in the review. The additional patients included strengthen the findings and conclusions of the study.

However the authors need to be more cautious about NAC being able to clear the aggregates from the vascular wall. The blood-brain barrier is at the level of the endothelium, which is the inner layer of the vessel wall. If NAC works as a "pipe cleaner", as stated, then it must be assumed that the aggregates cross the endothelium from the brain side leaving the vessel wall otherwise intact. The aggregates permeate the entire vascular wall and induce considerable structural and functional damage to the vessel wall through inflammation and oxidative stress, and removing them may still leave a damaged vessel behind. On the other hand, preventive treatment with NAC may protect the vessel through a "sink effect" draining low order aggregates into the circulation and preventing their accumulation in the vessel wall and consequent damage.

We thank the reviewer for these comments. The indicated text has been removed, and remaining text has been made more conservative regarding therapeutic effects and potential mechanisms, which remain to be explored.